# Citrulline Supplementation Improves Microvascular Function and Muscle Strength in Middle-Aged and Older Adults with Type 2 Diabetes

**DOI:** 10.3390/nu17172790

**Published:** 2025-08-28

**Authors:** Arturo Figueroa, Katherine N. Dillon, Danielle E. Levitt, Yejin Kang

**Affiliations:** 1Department of Kinesiology and Sport Management, Texas Tech University, Lubbock, TX 79409, USA; kdillon@albion.edu (K.N.D.); danielle.levitt@ttu.edu (D.E.L.); yejin.kang@ubc.ca (Y.K.); 2Department of Kinesiology, Albion College, Albion, MI 49224, USA; 3Department of Anesthesiology, Pharmacology & Therapeutics, University of British Columbia (UBC), Vancouver, BC V6T 1Z3, Canada

**Keywords:** vascular aging, peripheral artery tonometry, muscle tissue oxygen saturation, reactive hyperemia index, arginine precursor

## Abstract

Background: Patients with type 2 diabetes (T2D) develop vascular complications due to arginine deficiency-induced microvascular endothelial dysfunction, which is related to the loss of muscle strength (MS) associated with aging. Thus, increased nitric oxide (NO)-mediated vasodilation may improve MS. We investigated the impact of the NO precursor citrulline on microvascular function (endothelial and muscle reactivity) and MS in T2D patients. Methods: Sixteen participants with T2D (53–72 years, nine females) were randomized to citrulline supplementation (CITS, 6 g/day) or placebo for 4 weeks prior to an 8-week washout period, followed by the opposite supplement for 4 weeks in a crossover trial. Endothelial function (log-transformed reactive hyperemia index, LnRHI), forearm muscle reactivity (near-infrared spectroscopy-derived tissue oxygen index (TOI) reperfusion indices), plasma arginine levels (ARG), and handgrip strength (HGS_rel_) and calf MS (CMS_rel_) adjusted for body weight were measured at baseline and 4 weeks for each condition. Results: CITS increased the LnRHI (∆0.11 ± 0.16 vs. ∆−0.08 ± 0.24, *p* < 0.05), TOI range (∆2.6 ± 3.3 vs. ∆−1.5 ± 4.8%, *p* < 0.01), TOI hyperemic response (∆1.2 ± 1.4 vs. ∆−0.6 ± 2.8%, *p* < 0.05), TOI 2 min area under the curve (∆154 ± 187 vs. ∆−41 ± 194%/s, *p* < 0.01), ARG (∆43 ± 28 vs. ∆1 ± 16μM/L, *p* < 0.001), CMS (∆1.5 ± 2.8 vs. ∆−0.3 ± 1.2 kg, *p* < 0.05), and CMS_rel_ (∆0.02 ± 0.03 vs. ∆−0.01 ± 0.02 kg/kg, *p* < 0.01) compared to placebo. The improvements in LnRHI and CMS_rel_ were correlated (*r* = 0.37, *p* < 0.05). Conclusions: This study showed that CITS improves microvascular endothelial function, muscle microvascular reactivity, and calf muscle strength in middle-aged and older patients with T2D.

## 1. Introduction

The increased cardiovascular mortality risk in patients with type 2 diabetes (T2D) is associated with macrovascular endothelial dysfunction assessed using brachial artery flow-mediated dilation (FMD) [1]. However, patients with T2D have a greater incidence of microvascular than macrovascular complications [2]. Microvascular endothelial dysfunction precedes the progression of T2D [3] and its corresponding microvascular complications [4].

An overlooked microvascular complication of T2D is sarcopenia [5], defined as the loss of muscle function (strength and walking capacity) and mass associated with aging and accelerated by T2D [6]. Older patients with T2D have a rapid decline in leg muscle strength and walking capacity [6,7]. Although sarcopenia involves several pathophysiological mechanisms, growing evidence indicates that microvascular endothelial dysfunction may play a main role in decreased muscle function related to aging and T2D [8,9,10]. Microvascular endothelial dysfunction, measured using the finger reactive hyperemia (RH) index (RHI), is associated with low muscle strength (MS) and power [10,11], low muscle mass [5], and T2D [3,12]. Moreover, reduced skeletal muscle capillary perfusion and anabolic actions of insulin may lead to muscle weakness and slow walking speed [9,13,14]. Capillary reactivity in skeletal muscle can be assessed using near-infrared spectroscopy (NIRS)-derived tissue oxygen saturation index (TOI) indices following arterial occlusion (RH) [15]. These reperfusion indices are attenuated by aging [16,17,18] and impaired glycemic control [19,20]. However, studies evaluating muscle microvascular dysfunction using TOI reperfusion indices are lacking in patients with T2D.

Endothelial nitric oxide (NO) synthase (eNOS) produces NO via oxidation of the amino acid arginine. Impaired NO synthesis attenuates endothelial-mediated vasodilation, leading to endothelial dysfunction [3,21]. Arginine deficiency induced by arginase overactivation is a main mechanism behind microvascular endothelial dysfunction in T2D [22,23]. Competing with eNOS for the same substrate, arginase converts arginine to ornithine and urea, leading to arginine deficiency, eNOS uncoupling, and reduced NO synthesis [24,25]. Indeed, older patients with T2D have low arginine levels in plasma [7] and skeletal muscle [26], suggesting that impaired arginine metabolism is involved in sarcopenia.

The amino acid citrulline is an effective arginine and NO precursor due to improved absorption [27], conversion to arginine [28], and an arginase inhibitory effect [29,30,31]. Previous studies have shown vascular and muscular benefits of citrulline supplementation in non-diabetic populations [32,33,34]. However, the effectiveness of citrulline supplementation to improve microvascular function and muscle strength has not been previously investigated in patients with T2D. The purpose of this study was to test the hypothesis that citrulline supplementation would improve microvascular function, assessed using finger RHI and TOI reperfusion indices. We also hypothesized that limb muscle strength would increase with citrulline supplementation in patients with T2D.

## 2. Materials and Methods

### 2.1. Study Participants

Middle-aged and older adults (53–72 years) with clinical diagnosis of T2D were recruited from the general community in Lubbock, TX via advertisements on social media and in newspapers. Inclusion criteria were (1) physician-diagnosed T2D at least 3 months before the study, (2) T2D treated and controlled with oral hypoglycemic medications and/or insulin, (3) resting systolic blood pressure (BP) < 160 mmHg, and (4) body mass index (BMI) < 40 kg/m^2^. All women were post-menopausal defined as >1 year without menstruation. Exclusion criteria were (1) type 1 diabetes, (2) uncontrolled T2D (fasting blood glucose > 200 mg/dL) or uncontrolled hypertension, (3) other chronic diseases, (4) smoking, (5) excessive alcohol intake (>7 standard drinks per week for women and >14 per week for men), (6) consuming arginine- or citrulline-rich foods and/or supplements, and (7) highly physically active (>120 min/week of structured exercise or physical activity of moderate or high intensity) within the last 6 months.

### 2.2. Study Design and Protocol

This study was a randomized, placebo-controlled, double-blind crossover trial. Randomization was performed by the principal investigator (AF), who was not involved in laboratory measurements.

On visit 1, initial participant eligibility was determined via phone prescreening. Pre-qualified individuals came to the laboratory for an onsite screening visit. After 20 min of rest in the supine position, brachial BP was measured at least twice using an automated oscillometric device (HEM-907XL; Omron Healthcare, Vernon Hill, IL, USA) and averaged if there was a <5 mmHg difference in systolic BP. A blood sample via finger stick was analyzed to assess fasting glucose (Contour Blood Glucose Monitor; Bayer, Leverkusen, Germany) and hemoglobin A1C (HbA1C) (A1CNow+; PTS Diagnostics, Whitestown, IN, USA). Thereafter, they signed an informed consent and completed a health history questionnaire. The protocols were approved by the Institutional Review Board of Texas Tech University (IRB2022-1056) following the standards set forth by the Declaration of Helsinki. This study was registered on ClinicalTrials.gov (NCT06016478).

On visits 2–5, blood samples were collected using venipuncture, and anthropometry, body composition (at visit 2 only), vascular function, and MS were measured. Following visit 2, participants were randomized 1:1 to citrulline (6 g/day) or placebo (microcrystalline cellulose) (NOW Foods, Bloomingdale, IL, USA) for 4 weeks. During the 4-week supplementation period, participants took 4 × 750 mg capsules in the morning and 4 capsules in the evening one hour before or after their meals. The placebo and citrulline capsules were identical in size, shape, and color. Participants visited the laboratory again after the first supplementation period (visit 3). Then, after an 8-week washout period, participants crossed over to the opposite supplementation for 4 weeks (visits 4 and 5). Participants were asked to return unconsumed capsules on their laboratory visits 3 and 5 at the end of each 4-week period to assess adherence using capsule count. Participants were asked to maintain their medications, supplements, and dietary and physical activity habits until study completion. The last capsules for each condition were consumed 10–12 h before their laboratory visits at the end of each period.

### 2.3. Measurements

All laboratory measurements were conducted in the morning between 7–10 a.m. Participants had at least 8 h of overnight fast and refrained from medications, supplementation, food, and caffeine for at least 12 h, and from alcohol consumption and moderate to heavy physical activity for at least 24 h. Measurements were conducted in the same order on each occasion.

#### 2.3.1. Anthropometry, Lean Mass, and Muscle Strength

Height and weight were measured using a wall-mounted stadiometer and beam scale (Detecto, Webb City, MO, USA). BMI was calculated as body weight (kg) divided by height squared (m^2^). Waist circumference (WC) was measured with a non-elastic tape as a horizontal line at the midpoint between the lower rib and upper border of the iliac crest. Appendicular lean mass (ALM) was assessed using a whole-body dual-energy X-ray absorptiometry scan (GE Lunar DPX-IQ; Madison, WI, USA). Appendicular skeletal muscle mass index (ASMI) was calculated as ALM divided by height squared.

Maximum voluntary contraction of the dominant hand, a measure of handgrip strength (HGS), was evaluated 3 times, separated by 1 min of rest, in the standing position with the elbow at 90° using a handgrip dynamometer (Lafayette Instrument Co., Lafayette, IN, USA). The highest force was considered the maximum voluntary contraction. Calf MS was determined as the highest weight moved with proper form to complete 10 repetitions of plantarflexion exercise (10RM), obtained within 3–4 sets with 90 s of rest between sets, using a leg press machine (Cybex Eagle, Rosemont, IL, USA). Relative HGS (HGS_rel_) and calf MS (MS_rel_) were calculated as absolute strength divided by body weight in kg.

#### 2.3.2. Plasma Arginine Concentrations

A venous blood sample was collected, centrifuged, aliquoted, and stored in a −80 °C freezer for future analysis. A competitive-binding, colorimetric enzyme-linked immunosorbent assay (ELISA) was used to quantify circulating arginine following the manufacturer’s manual (IDK Arginine ELISA, Immundiagnostik AG, Bensheim, Germany). The intra-assay percent coefficient of variation (%CV) was ≤3.8%, and the inter-assay % CV, calculated from the inter-assay control, was 5.5%.

#### 2.3.3. Microvascular Function

Participants rested quietly in the supine position for at least 20 min prior to vascular measurements in a quiet, dimly lit, temperature-controlled room (23–25 °C). Microvascular endothelial function was assessed using peripheral arterial tonometry (PAT) in finger vessels and microvascular reactivity using NIRS in the forearm. For both microvascular measurements, a rapidly inflatable cuff was placed on the right forearm distal to the antecubital fossa. After baseline measurements, the cuff was inflated to 250 mmHg for 5 min (E20 Rapid Cuff Inflator, Hokanson, Bellevue, WA, USA). Thereafter, the cuff was rapidly deflated to induce RH for 3 min. Data were collected continuously during baseline, occlusion, and reperfusion periods.

##### Peripheral Artery Tonometry

PAT pneumatic probes were placed on the index fingers and connected to the device (EndoPAT-2000, Itamar Medical, Caesarea, Israel). The left index finger served as the control. PAT signals were collected continuously during 5 min baseline, 5 min occlusion, and 5 min reperfusion. Finger endothelial function was determined as the RHI, which is automatically calculated as the ratio of the PAT amplitude post-occlusion (A) to baseline (B) of the occluded arm divided by the PAT amplitude post-occlusion (C) to baseline (D) of the control arm and normalized to the baseline of the control arm: (A/B)/(C/D) × baseline. The natural logarithm-transformed RHI (LnRHI) was calculated as it is associated with T2D and obesity [12].

##### Muscle Microvascular Reactivity

Microvascular reactivity was assessed using NIRS-derived hemoglobin + myoglobin (NIRO-2000, Hamamatsu, Japan). The NIRS probe was placed longitudinally on the right flexor digitorum profundus. The source and sensor were separated by 4 cm, providing a 2 cm depth for tissue sample volume. The probe was secured and covered with a black cloth. The NIRS signals were continuously recorded at a frequency of 10 Hz during the 2 min baseline, 5 min occlusion, and 3 min reperfusion periods. TOI parameters used in the study were as follows: (1) baseline average (TOI_baseline_); (2) occlusion magnitude, the difference between baseline and minimum during occlusion (TOI_occmag_); (3) reperfusion magnitude (range), the difference between minimum and maximum during reperfusion (TOI_range_); (4) hyperemic response, the difference between the baseline and maximum (TOI_hyperesp_); and (5) area under the curve above the baseline during the first 2 min (TOI_AUC2min_). In the present study, TOI_range_, TOI_hyperesp_, and TOI_AUC2min_ were considered the main microvascular reactivity indices [16,17,18,35].

### 2.4. Statistical Analysis

Sample size was estimated a priori using G*Power (v 3.1.9.7, Dusseldorf, Germany) based on data that showed increased finger RHI after improved NO availability in T2D patients [36]. It was estimated that 16 participants would be required to detect a difference between groups with ≥80% power at the α = 0.05 level. Statistical analyses were performed using SPSS v. 29.0 (IBM SPSS Inc., Chicago, IL, USA). Normality was confirmed with the Shapiro–Wilk test and QQ plots. A two-way analysis of variance (ANOVA) with repeated measures was performed to determine significant differences in microvascular function, arginine levels, and muscle strength between groups (CIT vs. placebo) over time (pre vs. post). When significant time × group interactions were found, Bonferroni-adjusted post-hoc tests were used to determine group differences. Pearson’s correlation coefficients were used to assess the relationships between changes in microvascular function (RHI, LnRHI, and TOI-reperfusion indices) and changes in muscle strength. Data are presented as mean ± SD. Statistical significance was set at *p* < 0.05.

## 3. Results

### 3.1. Participants

A total of 118 individuals was screened for study eligibility (Figure 1). Twenty-two participants were initially randomized to either citrulline or placebo supplementation. One individual withdrew from each group during the first 4-week supplementation period. After an 8-week washout period, 10 individuals per group were allocated to the opposite supplementation (citrulline or placebo) group for the subsequent 4 weeks. Only 16 participants completed both supplementation periods and were considered in the final analysis. Compliance to citrulline and placebo was 92 ± 6% and 91 ± 7%, respectively. The supplements were well tolerated with no adverse effects reported, except one, who reported mild stomach discomfort.

Participant characteristics and medications are presented in Table 1. Metformin was the predominant medication (*n* = 13). None of the participants met the criteria for low HGS (HGS < 16 and <27 kg) and ASMI (<5.5 and 7.0 kg/m^2^) for women and men, respectively [37].

### 3.2. Anthropometry, Arginine Levels, and Muscle Strength

The body weight, BMI, arginine, HGS, HGS_rel_, calf MS, and calf MS_rel_ are presented in Table 2. No significant (*p* = 0.08) time × group interaction was observed for body weight. A significant time × group interaction was observed for BMI (*p* < 0.05). The decrease in BMI after citrulline supplementation was not significant (*p* = 0.08), while no significant changes were observed after placebo (*p* > 0.05).

A time × group interaction was observed for plasma arginine levels (*p* < 0.001), calf MS (*p* < 0.05), and MS_rel_ (*p* < 0.01), but not for HGS and HGS_rel_ in Table 2 and Figure 2. Citrulline supplementation significantly increased plasma arginine levels (∆43.3 ± 28.4 μM/L, *p* < 0.001), calf MS (∆1.51 ± 2.80 kg, *p* < 0.05), and MS_rel_ (∆0.02 ± 0.03 kg/kg, *p* < 0.01), while no significant changes were observed after placebo (*p* > 0.05).

### 3.3. Peripheral Artery Tonometry

There were significant time × group interaction effects for RHI (*p* < 0.05) and LnRHI (*p* < 0.05) (Table 3 and Figure 3). RHI (∆0.28 ± 0.41, *p* < 0.05) and LnRHI (∆0.11 ± 0.17, *p* < 0.05) increased after citrulline supplementation. The increase in LnRHI was significantly correlated with changes in calf MS_rel_ (*r* = 0.37, *p* < 0.05) (Figure 4). There were no significant changes in RHI and LnRHI after placebo (*p* > 0.05).

### 3.4. Muscle Microvascular Reactivity

There were significant time × group interaction effects for TOI_range_ (*p* < 0.01), TOI_hyperesp_ (*p* < 0.05), and TOI_AUC2min_ (*p* < 0.01) (Table 3 and Figure 5). TOI_range_ (∆2.60 ± 3.28%, *p* < 0.05), TOI_hyperesp_ (∆1.15 ± 1.38%, *p* < 0.05), and TOI_AUC2min_ (∆154 ± 187%/s, *p* < 0.01) increased after citrulline supplementation, while there were no significant changes with placebo.

## 4. Discussion

Our study is the first randomized, double-blind, placebo-controlled clinical trial to investigate the efficacy of citrulline supplementation on microvascular function and MS in patients with T2D. The main findings are that citrulline supplementation for 4 weeks improved microvascular endothelial function and forearm muscle TOI reperfusion indices compared to placebo. In addition, citrulline supplementation increased the plasma arginine levels and calf MS. The improvement in the LnRHI was related to the increase in the calf MS_rel_. Citrulline supplementation did not significantly improve the HGS.

A Low RHI is associated with T2D, obesity [12,38], and poor glycemic control [39]. At baseline, our participants’ mean RHI value was similar to that previously reported in newly diagnosed, well-controlled patients with T2D [36]. This study demonstrated that 4 weeks of citrulline supplementation increased the finger RHI by 13% in patients with T2D. Indeed, the RHI increased to a mean value previously reported in healthy controls [38]. The benefit of citrulline supplementation observed in the present study is important, since a low RHI can predict future cardiovascular events in T2D patients [39]. Consistent with our present findings, two previous studies examined the efficacy of citrulline on finger endothelial function. Citrulline supplementation for 2 weeks significantly increased the finger RHI by 19% in children and adolescents with mitochondrial diseases and endothelial dysfunction (RHI < 1.67) [40]. Although endothelial function was not assessed as RHI, citrulline for 2 months effectively increased the finger photoplethysmographic wave maximal amplitude time in adults with heart failure with preserved ejection fraction (HFpEF) [41], a population with peripheral endothelial dysfunction [34]. These findings suggest that citrulline is an effective NO precursor for improving microvascular endothelial function in clinical populations. Other studies have investigated the impact of increased NO bioavailability on finger RHI using NO enhancers (phosphodiesterase-5 [PDE5] inhibitors) or NO donors (nitrates) with contradictory findings. Endothelium-derived NO activates guanylate cyclase, converting guanosine triphosphate to cyclic guanosine monophosphate (cGMP) in vascular smooth muscle cells, leading to relaxation and vasodilation. Subsequently, NO is deactivated by PDE5; thus, PDFE5 inhibition is clinically used to prolong NO bioavailability. In a placebo-controlled trial, the effect of the PDE5 inhibitor tadalafil for 6 weeks on the finger RHI was examined in recently diagnosed, well-controlled patients with T2D [36]. The improved finger RHI following tadalafil treatment was attributed to increased NO bioavailability due to reduced cGMP degradation and prolonged action [36]. In agreement with the previous study, the mean improvement after tadalafil (∆0.30 [0.08; 0.52]) was similar to the change observed after citrulline (∆0.28 [0.06; 0.50]) in the present study. In contrast, beetroot supplementation increases NO bioavailability via the nitrate–nitrite–NO pathway, which does not involve the endothelium. In two recent studies, dietary inorganic nitrate (from beet root) for 4 weeks did not improve the finger RHI despite increases in plasma NO levels in middle-aged and older adults without cardiovascular or metabolic diseases [42,43]. A potential explanation for the inefficiency of the nitrate supplementation may be that elevated extra-endothelial NO may inhibit eNOS activity [44,45]. In contrast, citrulline activates the arginine–NO–cGMP pathway in small arteries [46]. In the present study, citrulline induced a beneficial vascular effect by increasing microvascular endothelial function in patients with T2D.

We assessed skeletal muscle microvascular reactivity to hyperemia using NIRS-derived TOI reperfusion indices, which primarily reflect increased capillary perfusion [15]. Muscle TOI_range_ [16,17] and TOI_AUC2min_ [18,35] are attenuated by aging, hyperglycemia [19], and obesity [47]. Our present data showed that citrulline supplementation for 4 weeks improved forearm muscle microvascular reactivity, assessed as TOI reperfusion indices, in patients with T2D. Only one study has investigated the efficacy of citrulline supplementation on muscle microvascular reactivity. In disagreement with the present findings, 6 g/day citrulline combined with nitrate supplementation for one month did not increase the muscle TOI_range_ in healthy older adults [48]. Moreover, a recent review article supports the ineffectiveness of acute ingestion and short-term dietary nitrate supplementation for improving microvascular reactivity in healthy populations [49]. Similarly, 6 g/day citrulline supplementation failed to improve brachial artery FMD despite significant increases in plasma arginine in healthy adults [50,51]. Therefore, it is possible that previous studies failed to observe beneficial vascular effects of NO donors and precursors in populations with normal endothelial function. Although short-term arginine supplementation has improved forearm muscle reactivity assessed using venous occlusion plethysmography and plasma NO levels in individuals with metabolic syndrome and prediabetes [52], long-term arginine supplementation becomes ineffective for vascular benefits in patients with cardiometabolic risk factors or diseases [53]. This ineffectiveness of chronic arginine supplementation is explained by stimulation of arginase activity, leading to reduced arginine bioavailability for NO synthesis [31]. Unlike arginine, citrulline is better absorbed, not metabolized by arginase, not extracted by the liver, converted to arginine in the kidneys [27,28], and, subsequently, converted to NO in endothelial cells [29]. In agreement with the present findings, Ratchford et al. [34] demonstrated in patients with HFpEF that 6 g/day citrulline for 7 days increased brachial artery hyperemic blood flow, which primarily measures forearm microvascular function via the arginine–NO pathway [54]. To the best of our knowledge, the present study is the first to show the effectiveness of short-term citrulline supplementation on muscle microvascular reactivity assessed as TOI reperfusion indices.

Arginine is the common substrate for the enzymes eNOS and arginase. Patients with T2D have low plasma arginine levels, a high ornithine/arginine ratio, and forearm microvascular dysfunction [55] due to augmented arginase expression and activity [22,31,56]. Increased arginase activity leads to arginine deficiency by converting arginine to ornithine and urea [22,55,57,58]. Although pharmacological arginase inhibition using intra-arterial infusion has been effective to increase forearm microvascular (muscle and skin) dilation in T2D patients [22,59], intra-vascular drug administration is not a practical strategy for treating endothelial dysfunction. Importantly, oral citrulline has an arginase-inhibitory effect that improved NO synthesis and endothelial-dependent vasodilation in diabetic rats [30,31]. In patients with T2D, supplementation with 2 g/day citrulline for 4 weeks increased plasma NO bioavailability via arginase inhibition [29]. However, endothelial function was not assessed in the previous study. In the present study, plasma arginine levels increased by 69% following 4-week citrulline supplementation in patients with T2D. This is in part due to a high absorption in the small intestine and conversion to arginine in the kidneys [28]. We and others have shown increases in circulating arginine levels with an improvement in macrovascular endothelial-mediated vasodilation (brachial artery FMD) following short-term (1–4 week) citrulline supplementation in clinical populations [32,34,50]. Conversely, citrulline supplementation was ineffective at decreasing circulating asymmetric dimethylarginine levels [50,51,55], an arginine analog that competes with arginine for eNOS binding. It appears that increased arginine bioavailability to endothelial cells concurrent with arginase inhibition is necessary to improve endothelial-mediated vasodilation [24]. Therefore, increased circulating arginine levels and NO-mediated vasodilation would explain the improvements in endothelial function in previous and present studies.

Although muscle mass is important for glucose uptake and metabolism, low HGS and walking performance are recognized as main indicators of sarcopenia [37]. In the present study, the increase in HGS_rel_ was not statistically significant (*p* = 0.09). The inefficacy of citrulline supplementation would be explained by the preserved normal HGS, as our participants had forearm strength levels above cutoffs for weakness (<16 and <27 kg for women and men) [37]. As T2D does not amplify the effect of aging on HGS [60], it may take longer supplementation periods to observe an improvement. In agreement with our findings, no improvement in HGS was observed in healthy adults aged 60–73 years following citrulline (3 g/day) malate for 6 weeks [61]. In contrast, walking speed improved in the study by Caballero-Garcia et al. [61], suggesting the effectiveness of citrulline to improve leg but not forearm muscle function. Similarly, a recent study showed improved gait speed in patients with HFpEF, a population with endothelial dysfunction and impaired functional capacity, after 1 week of citrulline supplementation (6 g/day) [34]. The previous studies provided evidence of citrulline effectiveness to improve walking capacity. However, walking speed is negatively affected by impaired calf MS in older adults with T2D [60], which was not measured in previous studies. Interestingly, we found that citrulline supplementation increased the calf MS_rel_ in middle-aged and older adults with T2D. In a previous study, we observed an increase in knee flexor MS but not in lean mass following citrulline supplementation (6 g/day) for 4 weeks in post-menopausal women [33]. In food-restricted adult female rats, the placebo group experienced a loss of muscle strength, whereas citrulline administration preserved leg muscle function without a significant effect on mass [62]. A potential mechanism underlying leg muscle dysfunction in older adults with T2D could be microvascular endothelial dysfunction [11,63]. Similar to those with T2D, middle-aged and older patients with chronic kidney disease have microvascular endothelial dysfunction associated with impaired walking speed and leg muscle weakness, with less impact on HGS [9]. Indeed, evidence of low arginine levels in plasma [7] and skeletal muscles [26] suggests reduced microvascular perfusion and delivery of amino acids may be involved in the decline of leg muscle function in middle-aged and older adults with chronic diseases. As an arginine precursor, citrulline has improved femoral artery endothelial function [33], leg muscle strength [33], and walking performance [34] in older individuals. In addition, citrulline-derived arginine is a powerful activator of protein synthesis, which improves calf MS [62]. We observed that the increase in calf MS_rel_ was related to improvements in finger LnRHI (*r* = 0.37, *p* < 0.05). These findings suggest that improved microvascular endothelial function may lead to greater calf MS after citrulline supplementation in patients with T2D.

The strengths of our study include the crossover design, the use of two techniques of microvascular function, and a long washout period. There are also several limitations to the present study. The sample size was small for evaluating sex differences. Participants were using medications with effects on blood glucose and blood pressure control. In agreement with previous studies, participants with chronic diseases using more than one medication had potential benefits on endothelial function (e.g., metformin, angiotensin-converting enzyme (ACE) inhibitors, calcium channel blockers, statins) [34,64]. Although insulin treatment may improve FMD in poorly controlled T2D patients [65], two of the three participants on insulin therapy in the present study had improved finger RHI and TOI reperfusion below the mean at baseline, while the other participant did not respond to citrulline supplementation. Despite medications being interrupted overnight, some persistent effects cannot be excluded. We did not measure changes in lean mass based on our previous findings of no improvements in appendicular lean mass after 4 weeks of citrulline supplementation in post-menopausal women [33]. Since in previous studies most of the participants were women [34,61], future studies should include equal number of men and women to examine sex differences in the effects of long-term citrulline supplementation.

## 5. Conclusions

Citrulline supplementation for 4 weeks significantly increased microvascular endothelial function and skeletal muscle microvascular reactivity in patients with T2D. The increase in microvascular endothelial function was correlated with an increase in calf muscle strength adjusted to body weight. Therefore, citrulline supplementation may be considered as an adjunctive non-pharmacological option in the management of vascular and skeletal muscle dysfunctions in middle-aged and older adults with T2D.

## Figures and Tables

**Figure 1 nutrients-17-02790-f001:**
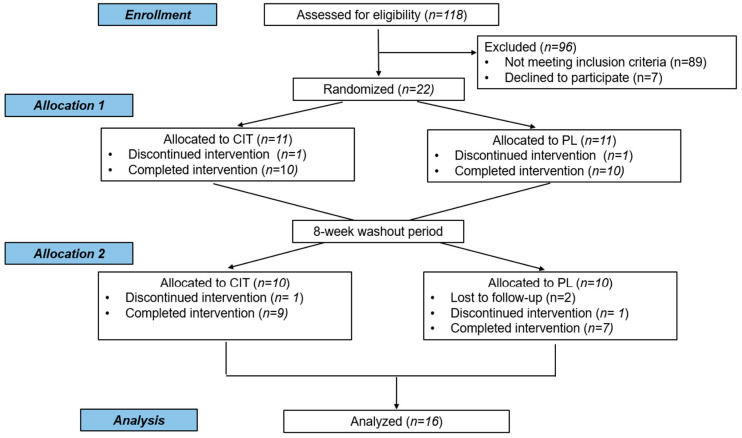
CONSORT flow chart of participants through the study. CIT, citrulline; PL, placebo.

**Figure 2 nutrients-17-02790-f002:**
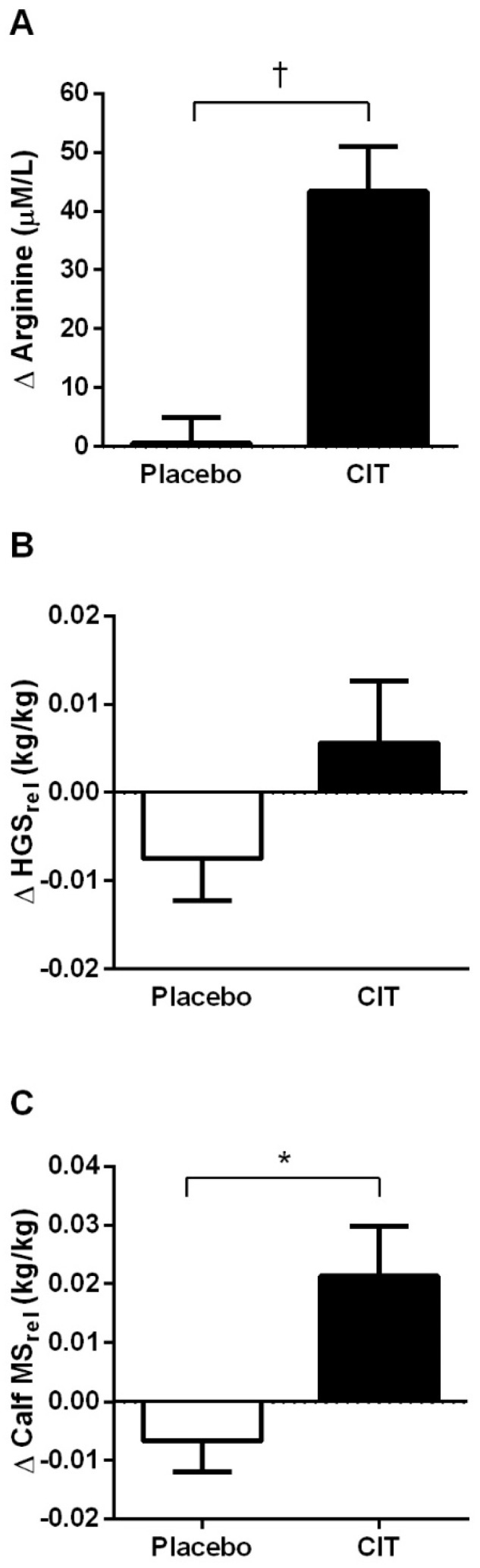
Changes (Δ) in plasma arginine levels (**A**), handgrip strength (**B**), and calf muscle strength (**C**) from 0 to 4 weeks. Abbreviations: CIT, citrulline; HGS_rel_, handgrip strength relative to body weight; MS_rel_, muscle strength relative to body weight. Values are mean ± standard deviation. * *p* < 0.01 and ^†^
*p* < 0.001 vs. placebo.

**Figure 3 nutrients-17-02790-f003:**
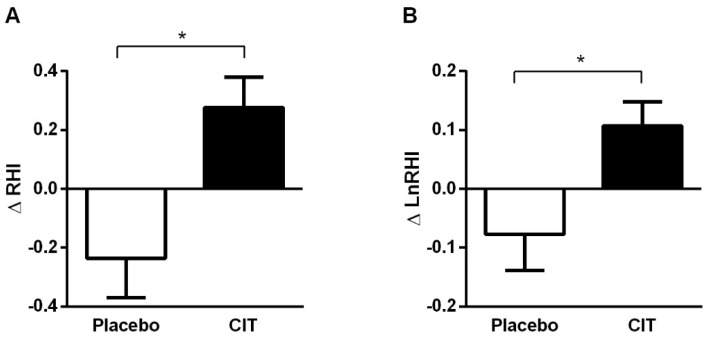
Changes (Δ) in finger reactive hyperemia index (**A**) and its natural logarithm (**B**) from 0 to 4 weeks. Abbreviations: CIT, citrulline; RHI, reactive hyperemia index; LnRHI, natural logarithm RHI. Values are mean ± standard deviation. * *p* < 0.05 vs. placebo.

**Figure 4 nutrients-17-02790-f004:**
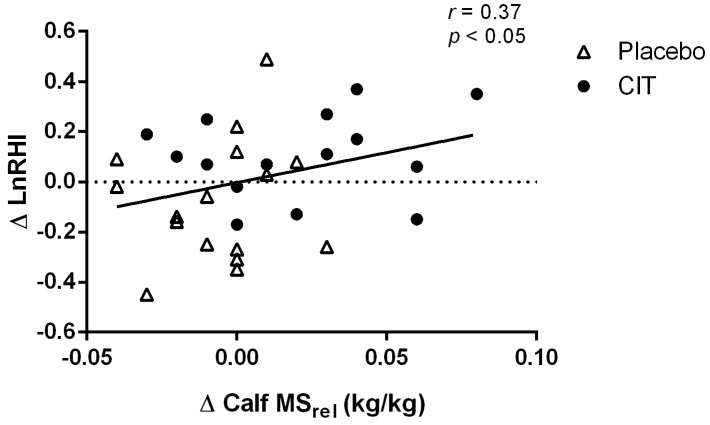
Relationship between changes in microvascular endothelial function (∆LnRHI) and calf muscle strength relative to body weight (∆ Calf MS_rel_) from 0 to 4 weeks. Relationship was determined by a Pearson correlation coefficient. Abbreviations: LnRHI, natural logarithm of the reactive hyperemia index; MS_rel_, muscle strength relative to body weight.

**Figure 5 nutrients-17-02790-f005:**
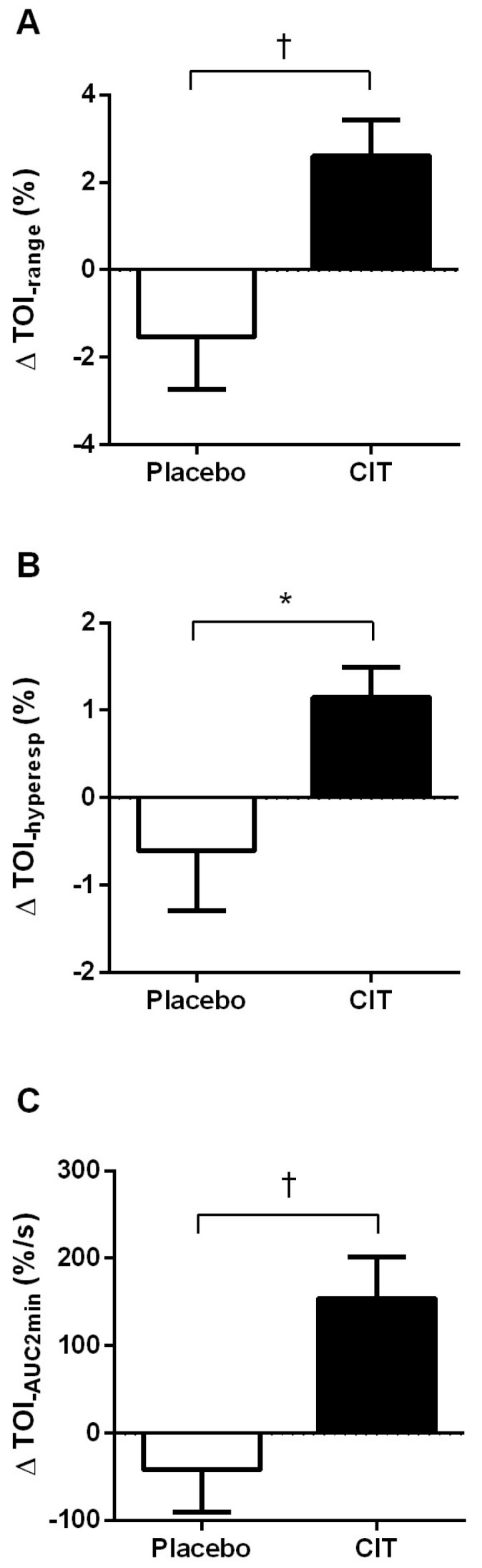
Changes (Δ) in forearm muscle TOI_range_ (**A**), TOI_hyperesp_ (**B**), and TOI_2minAUC_ (**C**) from 0 to 4 weeks. Abbreviations: CIT, citrulline; TOI, tissue oxygen saturation index; hyeperesp, hyperemic response; AUC2min, first 2 min of area under the reperfusion curve above baseline. Values are mean ± standard deviation. * *p* < 0.05 and **^†^**
*p* < 0.01 vs. placebo.

**Table 1 nutrients-17-02790-t001:** Participant characteristics from the screening visit.

	Value (*n* = 16)
Age (years)	62 ± 6
Sex (male/female)	7/9
Diabetes duration (years)	8 ± 5
Height (m)	1.66 ± 0.12
Weight (kg)	93.0 ± 18.7
BMI (kg/m^2^)	33.4 ± 4.9
Waist circumference (cm)	114.4 ± 11.6
Brachial SBP (mmHg)	130 ± 12
Fasting blood glucose (mg/dL)	139 ± 29
HbA1C (mmol/mol)	44 ± 8
HbA1C (%)	6.3 ± 0.8
HGS (kg)	32.2 ± 7.7
ASMI (kgm^2^)	8.1 ± 1.5
Hypoglycemic treatment, n (%)	
Metformin	13 (81)
Sulfonylureas	5 (31)
SGLT2 inhibitor	1 (6)
GLP-1 receptor agonist	4 (25)
Thiazolidinedione	1 (6)
Insulin	3 (19)
Anti-hypertensive medications, *n* (%)	
ACE inhibitor	4 (25)
ARB	5 (31)
Calcium channel blocker	2 (13)
Diuretic	4 (25)
Statin	9 (56)
Hormone replacement therapy, *n* (%)	
Estradiol	1 (6)
Progesterone	1 (6)
Levothyroxine	1 (6)

Values are mean ± standard deviation or number of participants (%). Abbreviations: BMI, body mass index; SBP, systolic blood pressure; HbA1C, hemoglobin A1C; HGS, handgrip strength; ASMI, appendicular skeletal muscle mass index; SGLT2, sodium-glucose co-transporter 2; GLP-1, glucagon-like peptide 1; ACE, angiotensin-converting enzyme; ARB, angiotensin II receptor blocker.

**Table 2 nutrients-17-02790-t002:** Anthropometry, plasma arginine levels, and muscle strength at pre- and post-intervention.

	Citrulline (*n* = 16)			Placebo (*n* = 16)			
	Pre	Post	*p* ^b^	Pre	Post	*p* ^b^	*p* ^a^
Height (m)	1.66 ± 0.12	-		1.66 ± 0.12	-		
Weight (kg)	93 ± 18	92 ± 18	0.15	92 ± 19	93 ± 18	0.30	0.08
BMI (kg/m^2^)	33.4 ± 5.1	33.2 ± 4.9	0.08	33.1 ± 4.8	33.3 ± 4.9	0.26	0.047
L-arginine (μM/L)	77 ± 19	120 ± 30	0.001	83 ± 18	84 ± 19	0.93	0.001
HGS (kg)	32 ± 8	33 ± 8	0.38	32 ± 8	32 ± 8	0.26	0.16
HGS_rel_ (kg/kg)	0.35 ± 0.09	0.36 ± 0.09	0.31	0.36 ± 0.10	0.35 ± 0.09	0.16	0.09
Calf MS (kg)	99 ± 26	100 ± 26	0.01	102 ± 27	101 ± 27	0.59	0.03
Calf MS_rel_ (kg/kg)	1.07 ± 0.22	1.09 ± 0.21	0.004	1.11 ± 0.23	1.11 ± 0.23	0.28	0.006

Values are mean ± standard deviation. Abbreviations: BMI, body mass index; HGS, handgrip strength; HGS_rel_, HGS relative to body weight; MS, muscle strength; MS_rel_, MS relative to body weight. *p*^a^-values are the time × group interaction from two-way repeated measures ANOVA. *p*^b^-values for within-group differences.

**Table 3 nutrients-17-02790-t003:** Microvascular function at pre- and post-intervention.

	Citrulline (*n* = 16)			Placebo (*n* = 16)			
	Pre	Post	*p* ^b^	Pre	Post	*p* ^b^	*p* ^a^
RHI	2.33 ± 0.30	2.60 ± 0.46	0.02	2.43 ± 0.47	2.26 ± 0.51	0.16	0.01
LnRHI	0.84 ± 0.13	0.94 ± 0.17	0.05	0.87 ± 0.20	0.79 ± 0.22	0.15	0.01
TOI-baseline (%)	71 ± 4	70 ± 4	0.06	71 ± 4	71 ± 4	0.78	0.20
TOI-_occmag_ (%)	−18 ± 7	−19 ± 8	0.08	−18 ± 8	−17 ± 7	0.33	0.06
TOI-_range_ (%)	25 ± 10	27 ± 11	0.02	25 ± 11	23 ± 9	0.15	0.008
TOI-_hyperesp_ (%)	7.1 ± 3.4	8.2 ± 3.4	0.04	7.0 ± 4.0	6.4 ± 3.0	0.28	0.03
TOI-_AUC2min_ (%/s)	604 ± 355	757± 415	0.003	582 ± 345	540 ± 259	0.39	0.007

Values are mean ± standard deviation. Abbreviations: RHI, reactive hyperemia index; LnRHI, natural logarithm of RHI; TOI, tissue oxygenation index; _occmag_, occlusion magnitude; _range_, reperfusion magnitude; _hyperesp_, hyperemic response; _AUC2min_, 2 min area under the curve after return to baseline. *p*^a^-values are the time × group interaction from two-way repeated measures ANOVA. *p*^b^-values for within-group differences.

## Data Availability

The original contributions presented in the study are included in the article. Data can be made available upon request to the corresponding author.

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
