# Peer review of "Citrulline Supplementation Improves Microvascular Function and Muscle Strength in Middle-Aged and Older Adults with Type 2 Diabetes"

_nutrients, 2025, doi:10.3390/nu17172790_

Round 1
Reviewer 1 Report
Comments and Suggestions for Authors
Minor revisions and considerations are as follows:
- The sample size is too small; please acknowledge this as a limitation of the study.
- Please check the number of female participants reported in the abstract, as it does not match the number indicated in Table 1.
- Please verify the age range reported in the abstract and ensure it aligns with the data presented in Table 1.
- Please double-check the formatting of the references.
Author Response
Responses to reviewers
Reviewer #1
We appreciate your time and effort invested in the review of our manuscript.
English Language and Figures:
- The English could be improved to more clearly express the research.
Response: We reviewed the manuscripts and made some corrections throughout the text (highlighted in yellow).
- Figures and tables must be improved
Response: We made some changes to the footnotes in figures (2, 3, and 4) and tables (2 and 3). In addition, columns for within-group differences p-values were added in tables 2 and 3.
Minor revisions and considerations are as follows:
- The sample size is too small; please acknowledge this as a limitation of the study.
Response: we acknowledged the small sample size as a limitation.
- Please check the number of female participants reported in the abstract, as it does not match the number indicated in Table 1.
Response: It was a mistake. There were 9 women in this study.
- Please verify the age range reported in the abstract and ensure it aligns with the data presented in Table 1.
Response: age range and mean ± SD are correct.
- Please double-check the formatting of the references.
Response: We checked the formatting of references.
Reviewer 2 Report
Comments and Suggestions for Authors
Congratulations to the authors. They have addressed a very attractive and novel clinical issue of sarcopenia as a microvascular complication of diabetes. In a double-blind, placebo-controlled experiment, the authors have found that citrulline improves endothelial reactivity and muscle strength in diabetic patients.
Minor comment:
Check line 285: Is RIH mistakenly entered instead of RHI?
Author Response
Reviewer #2
Comments and Suggestions for Authors
Congratulations to the authors. They have addressed a very attractive and novel clinical issue of sarcopenia as a microvascular complication of diabetes. In a double-blind, placebo-controlled experiment, the authors have found that citrulline improves endothelial reactivity and muscle strength in diabetic patients.
Minor comment:
Check line 285: Is RIH mistakenly entered instead of RHI?
Response: Yes, it was a typo. It is RHI. We appreciate your positive comments.
Thank you.